# Emerging Challenges in Technical Vocational Education and Training of Pakistan in the Context of CPEC

Naila Bano [1,*], Siliu Yang [1] and Easar Alam [2]

1   School of Public Policy and Management, China University of Mining and Technology, Xuzhou 221116, China; yangsl@cumt.edu.cn
2   School of Environment and Spatial Informatics, China University of Mining and Technology, Xuzhou 221116, China; easaralam@gmail.com
*   Correspondence: nailaeasar88@gmail.com

**Abstract:** Pakistan is a country with rich natural and human resources. The role of highly skilled people in national development has become enormously vital in the new developmental period, but it is also an irrefutable fact that the gap in highly skilled personnel in Pakistan is expanding. The organization of Technical Vocational Education and Training was introduced to prepare a skilled workforce for various industries and sectors in Pakistan; however, the 60% level of young, unskilled, and semi-skilled labor emerging from informal and non-formal sectors is largely attributed to the failure of Technical Vocational Education and Training to supply the country with its requirements for trained manpower for the economy and China Pakistan Economic Corridor (CPEC) projects. China and Pakistan launched historic projects such as CPEC as part of the Belt and Road Initiative (BRI), which fostered economic cooperation and development between the two countries. This article will go through the overview and the course of Technical Vocational Education and Training (TVET) in Pakistan. The major purpose of this study is to highlight that TVET in general, and CPEC in particular, are suffering from a lack of qualified personnel because of a variety of other reasons, such as outdated equipment, a lack of industry connectivity, inadequate skills, unemployment, and so on. The study is descriptive and exploratory in nature, and it employs a qualitative research method. The perspectives of the TVET challenges in Pakistan were researched using the data obtained from 500 student and staff respondents, including teachers, TVET workers, and TVET job holders. Some of the important findings include the fact that the current state of the TVET institutions is no doubt due to infrastructural issues and a lack of funding. In addition, TVET in Pakistan is marked by inadequate skills, a lack of industry connectivity, unemployment, insufficient teacher training, and a lack of female participation. In this study, recommendations were given based on the research analysis and research findings.

**Keywords:** technical vocational education and training; CPEC; challenges; Pakistan

## 1. Introduction

A global development strategy sponsored by China, the Belt and Road Initiative (BRI), includes infrastructure development, energy, and investment in over 152 nations and international organizations. The BRI aspires to enrich numerous Asian countries (Benard 2020), and the globalization of Chinese enterprises has raised the demand for technical and qualified workers, posing new challenges for vocational education (Salman et al. 2019). The China–Pakistan Economic Corridor, a part of the BRI, contains two key components: a new commercial and transportation route from Kashgar to Gwadar and SEZs (special economic zones) along the route, with power plants and other facilities (Ahmad 2020). The CPEC would enhance per capita productivity and youth participation in Pakistan's disadvantaged regions, while reducing poverty and unemployment by almost one million jobs, if it was properly planned and aligned with the planning commission's goals and

programs. The twofold advantage of the CPEC can be reaped through a well-coordinated and result-oriented strategy that incorporates a wide range of projects. As part of the CPEC, Pakistan's TVET sector would be modernized through technology transfer and the training of Pakistani youth to the CPEC standards.

The correct human resource development policy is crucial for economic growth. Mello (Mello 2008), Heckman, and Lochner (Heckman et al. 2008) assessed the role of technology and educated labor in each country's economic progress. Under the hypothesis of human capital theory, investing in general and vocational education is vital in the building of human capital. Overall, a balanced approach to general education and skill development leads to stronger economic growth and development (UNESCO 2016). Vocational education research began in the 1880s, during a time of rapid urbanization, automation, and industry (Maclean and Wilson 2009b). The application of technical and vocational skills to the workplace is an essential area of labor and employment. Around 80% of all jobs are considered to be of this type (UNESCO 2006). It was not until 1999 that UNESCO adopted the name TVET (Karmel 2010); various labels were used in past, such as apprenticeship training, vocational education, industrial arts, technical education, technical/vocational education (TVE) in Europe, occupational education (OE), vocational education and training (VET), and career and technical education (CTE) in the United States (Rauner and Maclean 2008). For Finch and Crunkilton (1979), the term "TVET" means education and training that prepares people for employment and increases their efficiency in numerous economic domains. Maclean and Wilson's empirical research stressed the relevance of TVET programs for young people in obtaining job experience or becoming self-employed and earning a steady income (Maclean and Wilson 2009a). TVET clearly plays an important role in social growth and citizenship sustainability. Jallah (International, Unesco, and Experts Meeting 2004) and Goel (Goel et al. 2010) believe that TVET is a "master key" for long-term progress, and skills and expertise are vital components of every nation's social and economic prosperity. According to Ayonmike (Ayonmike et al. 2015), VET is essential for preparing workers to deal with fast technical innovations. Uwaifo examined the technical professionals who originate, assist, and implement technological innovation (Uwaifo 2010). A world study (The World Bank 2015) connected HRD to economic development by supplying technically qualified personnel at all levels to satisfy the socio-economic criteria for industrial growth and advancement, without which capital would be lost. Thus, TVET is a learning process that involves, in addition to general education, technological study and occupation-specific practical skills. The CPEC network, worth USD 62 billion, aimed to construct energy projects and SEZs to consume talented and semi-skilled workers from both nations (Ahmad and Sharif 2016; Ali 2016; Mamoon and Shield 2018). The expansion of numerous firms needs qualified, technically certified, and talented professionals. With the lack of TVET growth, and few professional vocational colleges and a lack of technical and skilled abilities, meeting the cooperation's requirement for a technical and skilled workforce is challenging. Because CPEC would result in the expansion of institutes of applied and engineering sciences, the HRD policy to expand the number of professionals, executives, managers, and competent technical employees is crucial. Unlike capital, human resources actively contribute to social and economic advancement (Parnes 1965; Magsi 2016; Wang et al. 2017).

With Chinese and international investment in unexplored Baluchistan, the CPEC helps both China and Pakistan with a positive impact on the economy, development, and regional ties; however, for socioeconomic advancement, significant TVET system reforms are required to increase technical expertise (GOP 2018; Ansari and Wu 2012; Khan et al. 2020). Only 6% of workers are qualified in technical and vocational trades, which is not enough to fulfill current market demands and serve the CPEC projects. According to Ejaz Hussain, Pakistani workers cannot use Chinese technology instruments and machinery (Hussain and Rao 2020). To the best our knowledge, no conceptual or empirical studies have been performed in Pakistan to examine the relevance and challenges of TVET in the light of the CPEC. A few scholars have worked on this topic, but they were mostly confined

to one province; so, there was a need for further study in this field. It was because of this gap that the study in this paper was conducted. This research paper was based on the objective of identifying the major factors affecting Pakistan's TVET and examining the relationship between these variables in the CPEC context. There are many factors affecting TVET; however, only the following factors will be observed in this study: learning and physical facilities, employability, female participation, skills, industrial linkage, and teacher training.

## 2. Literature Review

### 2.1. TVET and LPS

When appropriate workshop facilities are provided, students can practice and demonstrate their skills. Facilities and management aid education. Multidisciplinary support is both a friend and a foe to planners. Virtual and real-world environments impact student learning. Formal education is influenced by the physical and aesthetic elements of the learning environment (Amin et al. 2012). With regard to enhancing teaching and learning environments, recent research and comparable findings show that good facilities help learning, whereas inferior facilities inhibit student accomplishment (Audu et al. 2013; Mendell and Heath 2005; Hill and Epps 2010; Earthman 2002; Uline and Tschannen-Moran 2008). Technical skills can only be learned in a well-established workshop with the appropriate tools, equipment, and machines (Audu et al. 2013). Suleiman and Hussain (Suleman et al. 2014) emphasized a unique physical classroom setup to aid education and improve learning. Inadequate workshop facilities in Pakistan pose a challenge and hamper the ability of students to learn new skills. Akomolafe and Adesua (Akomolafe and Adesua 2016) found a relationship between student motivation and academic achievement in west Nigeria. According to their research, boosting financial and budgetary resources to improve educational facilities may encourage students. The study by Umer (Umar and Ma'aji 2010) stressed the importance of proper workshop facilities in implementing the TVET curriculum. Classrooms, libraries, recreational equipment, and other school structures for boosting academic achievements were identified as examples of facilities by (Alimi et al. 2012). This study is backed up by Akomolafe and Adesua (2016), who found that students were more motivated to learn and performed better at institutions with better physical amenities. A positive and statistical impact on trainee skill development was found in the Dorcas study on the physical facilities in the TVET institutes in the Lake Victoria region (Ojera 2021). Audu Rufai (Audu et al. 2013) affirmed that a training program where the equipment is not functional will not only suffer in production but will also produce unskilled and unemployed personnel. An examination-driven approach to training, coupled with inadequate instructional materials, high student enrollment, inadequate training facilities, and a lack of collaboration with local industries to provide hands-on experience for trainers and trainees led Dasmani (2011) to conclude that students are receiving ineffective and inefficient training.

### 2.2. TVET and Employability

Scholars, such as Malechwanzi and Wanjala (Musyimi et al. 2018), believe the TVET value is defined by how well students can establish their developed skills and employability, while others link the TVET value to how well institutions enable learning and the learners' growth. Quality in the TVET programs, according to Demessew, is linked to the declared goals or aims of TVET, which vary by country (Woldetsadik 2012). In other words, the amount of quality skills required by specific countries is dictated by their labor needs. According to a UNESCO report, TVET aims to prepare young graduates for employment by providing them with the relevant work-based learning experience, technical skills, and theoretical knowledge (Hodge 1992; Zarini and Karina 2004). In most countries, VET is towards the top of the national political agenda. New technologies are replacing jobs and changing skill needs. As a result, malleable workers are in high demand (Hodge 1992). Colley asserts that the effective usage of multimedia and cooperative learning designs in

education helps both students and teachers to enhance their learning skills. As a result, learning to use technology can help students prepare for a number of careers. Assisting vocational education with the integration of technology has been a popular trend (Wolf 2011; Colley et al. 2003).

The Leitch Review of Skills concluded that developing skills by firms, individuals, and governments has benefited and will continue to benefit them (Leitch 2006). In order to enter the information economy, Pakistan must first escape from the low-level skills pit that TVET institutions have created (Amjad 2005). Given the growing young population, the TVET sector's capacity to provide demand-driven training programs to boost the workforce's technical and professional skills is inadequate. Moreover, compared to other developed countries in the region, Pakistan's labor force participation rate is low (only 44 percent). As a result, the majority of the population is jobless, which affects health, education, and quality of life (Janjua and Irfan 2012).

### 2.3. TVET and Female Participation

Those with less academic ability and ambition are more likely to enroll in lower-level TVET programs, whereas those with better ability and desire are more likely to enroll in higher-level TVET programs (Curtis 2008). In a study in rural China on the relationship between household wealth, educational goals, and labor market outcomes, the researchers discovered that household assets matter more for girls than for boys (Deng and Sherraden 2012). Sadia's research also shows that financially stable parents have more options when choosing a school for their kids (Ashraf 2019). In the literature, the education of the parents is an important predictor of human capital investment decisions (Tansel 1998). There is a strong correlation between a father's education and his children's educational success; furthermore, parental education tends to have effects on boys' and girls' educational and training choices (Nancy 1982; Dostie 2006). According to Malik Muhammad's research, the sons of highly educated fathers are more likely to pursue further education than the sons of less-educated fathers; these inequalities are particularly pronounced in rural and urban settings (Muhammad and Jamil 2021). Similarly, parents with more schooling years send their kids to school more than parents with less schooling years (Siddiqui 2017). Whether in formal, informal, or technical education, female participation is always questioned. Female involvement in TVET institutions is influenced by government, school, and societal issues, which explains why in most schools and universities male students dominate engineering and science (Ayonmike 2014; Kilango and Qin 2017). Khan (Khan 1986) and Jyoti (2012) also stress the perspective, facilities, prejudice, and recognition of female participation. Despite advancements in development and female inclusion, a 50% gender disparity in labor force participation rates continues, according to World Bank Group research. Female involvement is improved by higher educational attainment, but this link has diminished over time and is no longer adequate to overcome other individual and household variables (Lopez-Acevedo 2021).

### 2.4. TVET and Skills

Apart from technical expertise and long-term employment, employment skills are the most in-demand abilities in the global industrial sector (Farah 2018). The economies of the 21st century have turned the skills into a global currency. However, as the labor market demand develops and individuals lose the skills they do not use, the currency may depreciate. Various studies have proved that TVET is the key to skill development. It is characterized by focusing on a specific occupation so that individuals can find jobs related to their skills or start their own jobs. According to the World Bank's jobs and productivity skills framework, the conclusion is that cognitive abilities, social behavior skills, and technical skills (a blend of cognitive and non-cognitive skills) are three types of skills (Ismail and Mohammed 2015). By instilling both cognitive and non-cognitive attributes in people, schooling, according to Heckman (Neumark et al. 2021) contributes to both earnings differentials and the development of acceptable policy options for human

capital construction. Higgins stressed that vocational education allows students to find and develop a wide range of talents, such as knowing what jobs are accessible and how to acquire the necessary skills. A solid transitional infrastructure for school leavers with few or no degrees is needed; yet, the respondents to this study felt they were "the Cinderella of the industry" (Higgins 2010). Therefore, it is critical for improved future prospects that the need for certain skills is met. As an example, the Saudi Arabian TVET does not have issues such as LDCs; yet, it is facing a shortage of skilled manpower in quantity and quality as well because of the devastating effects of labor market distortion and a bad social perception of skilled and manual labor in Saudi Arabia (Ryan 2001; Mellahi 2006).

A shortage of skilled workers also hinders the CPEC's excellence and timely completion, according to Magsi (2016). Ahmad discovered that the quality of the labor force is reliant on training, education, physique, and health. The quality of labor is important. The country's labor force is underperforming in terms of productivity due to the loss of human capital capacity (Personal and Archive 2016).The number of technical institutes and training centers in Pakistan is much smaller than the number of general education institutions in terms of enrollment potential, graduations, and the development of human capital. With TVET, countries such as Pakistan can expect higher producer profitability and salary increases and job prospects for qualified employees, as well as general economic growth, according to Kazmi (1990). In their research, (Mustafa et al. 2005) looked at the relationship between economic growth and skill development variability and discovered a strong correlation. Human capital growth and skill development, as well as other major macroeconomic metrics, ranked Pakistan worse than other South Asian countries in terms of competitiveness and global linkage (Khan et al. 2013; Khilji et al. 2012).

*2.5. TVET and IL*

Technical education shouldn't be as widely diffused. While the businesses lack competent workers, Pakistan's young unemployment rate continues to rise. Focusing on quality technical education can help narrow this gap (Ansari and Wu 2012). In Pakistan, skill development has been neglected. Pakistan has failed to develop cognitive, personal, and social capabilities. This reduces output, exports, and employment while raising living standards (Kemal 2005). Raimi's findings show that TVET has only a limited impact on employability and national growth because of influential elements such as funding, public perception, industrial linkage, and expertise (Raimi 2014). The primary client of TVET is industry. Unless there is a connection between TVET and industry, the graduates' professional skills will deteriorate and industrial output will stagnate. Those who graduated from TVET have gone on to work in a variety of nations (Komla and Offei-Ansah 2011). In most nations, there is a significant discrepancy between what students learn in the classroom and the real-world context in which they live and work. The TVET–industry ties are critical because they spur innovation both domestically and internationally. They also aid in the development of a competitive market and the economy. For TVET to be considered a success, it must be able to generate workers who are ready for the workforce when it is needed. The TVET–industry collaboration is essential. For graduates of technical schools, industry is a major source of employment, and technical schools are the primary source of training for the industry (Amalia and Stuart 2003). TVET in many countries can "stay stuck into the role of being a mere supplier of skilled labor to industry" as Majumdar (2011) points out; yet, the companies with connections to educational institutions have greater output rates. It is imperative that TVET universities collaborate with industry in order to meet the challenges posed by rapid technology advancements, new occupations, shifting employment requirements, and increased competitiveness (Uddin 2013). Vocational and technical education faces a slew of issues. The need for new talent necessitates that educational institutions provide not only the minimum of occupational training but also the training for scientists, inventors, and high-level professionals (Marope et al. 2015).

### 2.6. TVET and TT

According to Leke's research, vocational education and training (VET) subjects are less useful than academic subjects; on the other hand, the absence of learning and teaching facilities for the implementation of VET topics is a policy concern (Leke 2010). The variables that have the greatest impact on the delivery of high-quality vocational education and training in schools are, among other things, quality teacher training, physical resources, industry placement, TVET school/college management, and government support (Bartlett 2004). According to George, a common practice among TVET institutions is for level 3 pupils to be taught by junior teachers who are level 4 themselves. From a professional standpoint, this strategy is inefficient because they do not have the requisite abilities and because they have not been properly prepared (Spöttll 2009).

### 3. Conceptual Framework

The key issues are the main factors that influence the quality of TVET. These indicators are depicted in the framework given in Figure 1.

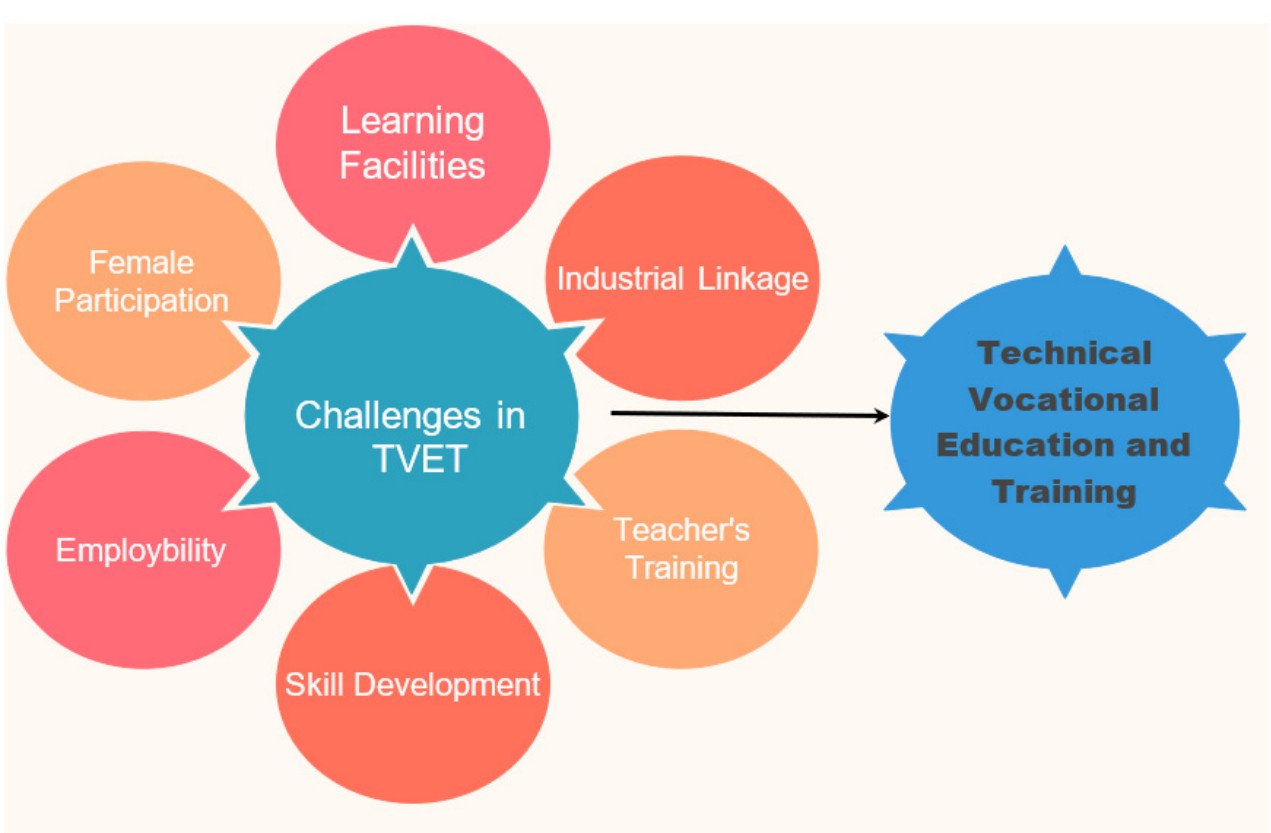

**Figure 1.** Conceptual Framework.

TVET is a dependent variable, but the rest of the variables are all independent. In this case, the additive model is applied. The following is the equation for the representation model:

$$Yi = \beta_0 + \beta_{xi} + \varepsilon_i \tag{1}$$

$Yi$ denotes the dependent variable, $\beta_0$ is the constant, $\beta$ is the regression coefficient of the independent variables, xi is the independent variables, also known as explanatory variables, and $\varepsilon_i$ denotes the random error. As a result, the equation that represents our conceptual framework is as follows:

$$Yi = \beta_0 + \beta_1 (LPS) + \beta_2 (EM) + \beta_3 (FP) + \beta_4 (LS) + \beta_5 (TT) + \beta_6 (IL) + \varepsilon_i \tag{2}$$

The dependent variable is represented by Y (TVET) Technical Vocational Education and Training, $\beta_1$ (LPS) is an independent variable and represents the learning and physical facilities. $\beta_2$ (EM) is an independent variable that represents employability, $\beta_3$ (FP) represents female participation as an independent variable, $\beta_4$ (LS) is skill development and represents an independent variable, $\beta_5$ (IL) is an independent variable that represents the industrial link, and $\beta_6$ (TT) represents the independent variable of teacher training.

The following hypotheses are generated on the basis of the conceptual framework.

**Hypothesis 1.** *Learning and physical facilities have a positive and significant impact on Technical Vocational Education and Training (TVET).*

**Hypothesis 2.** *Employment opportunities by CPEC have a positive and significant impact on Technical Vocational Education and Training (TVET).*

**Hypothesis 3.** *Female participation has a positive and significant impact on Technical Vocational Education and Training (TVET).*

**Hypothesis 4.** *Skills for self-employment have a positive and significant impact on Technical Vocational Education and Training (TVET).*

**Hypothesis 5.** *The industrial link has a positive and significant impact on Technical Vocational Education and Training (TVET).*

**Hypothesis 6.** *Teacher training has a positive and significant impact on Technical Vocational Education and Training (TVET).*

## 4. Research Methodology

### 4.1. Research Design

Researchers must rely on applicable literature and well-structured questionnaires when conducting their research. Questionnaires, interviews, and observations are often used to collect descriptive data (Mills and Gay 2011). This research was based on a survey. The survey research looks at the associations between variables in a subset of the population in order to extrapolate to the entire population (Pinsonneault and Kraemer 1993). There were six independent variables: learning and physical facilities (LPS), employability (EM), female participation (FP), skill development (LS), industrial linkage (IL), and teacher training (TT). The dependent variable of this study was Technical Vocational Education and Training.

### 4.2. Sample

According to McDaniel and Gates (McDaniel and Roger 2015), sampling is the method of collecting information from a subset and a sample of a large population. Purposive sampling was used in this study as only a particular section of the population possessed the relevant data. Non-probability sampling approaches are referred to by this methodology. The researchers employed this technique in this study to obtain data from people who were considered to be experts in their field. In this study, having this information was necessary for doing the quantitative research. The online survey was made as short and accessible as possible for the target population and the sample frame. The Web URL was sent to the intended participants (study sample) of the survey. Five hundred responses were used to compile the data out of an estimated total of 550 questionnaires; follow-up calls and emails to the groups were used to ensure the originality of the response, with a response rate of 90.9%.

### 4.3. Target Population

A population, according to Mendenhall (Stewart and Mendenhall 1990) is a group of people about whom the researcher wants to learn more. The target population involved in this study comprised the staff, including the teachers, technical job holders, entrepreneurs, and others, and the students of the government technical training centers, the Government College of Technology, the vocational training institutions, and the private technical institutes of Pakistan for boys and girls from the major cities Gilgit Baltistan, Islamabad, Rawalpindi, Lahore, Peshawar, Karachi and Faisalabad, Sialkot, and Sukkur. The numerical size of the subpopulation of students was 64%, and the staff was 36%, including teachers, technical job holders, entrepreneurs, and others (non-technical).

### 4.4. Data Collection

The questionnaire survey is a more effective method, because it takes less time, is cheaper, and can collect a large number of samples (Best and Khan 2006). It is well suited to collect standardized and quantifiable knowledge from all the members of the sample. It is simple to fill in and to keep the subjects on the subject; it is comparatively objective, and it is easy to tabulate and analyze (Bennett et al. 1984). The survey consisted of 36 items, and the items of the instrument were derived from various studies on the topic, such as those of Triki Nuri (Triki 2010), Dobrow (Dobrow and Tosti-Kharas 2012), Obiyo and Eze (2015) and Jamabo (2014). As the questionnaires were primarily based on the five-point Likert scale format style, most were requested to signify the degree of importance of each element in a list as Strongly Disagree, Disagree, Neutral, Agree, or Strongly Agree (Oppenheim 1992). In this research, a comprehensive questionnaire was developed online using a Google form, which comprised two parts. The first part included the demographic characteristics of the respondents, and the second descriptive part consisted of quantitative questions.

### 4.5. Data Analysis

This research was based on a survey. Examining the results of a survey is what the survey research entails. All the statistical methods were carried out using IBMS SPSS 26 to analyze the quantitative data. The Pearson Product–Moment Correlation (Kurtz and Mayo 1978) was performed to examine the probable correlations between the various variables. The important factors were identified using regression analysis. Multiple regression allows a researcher to derive a smaller collection of variables from a larger number of predictors by removing extraneous predictors, simplifying the data, and improving the predicted accuracy (Halinski and Feldt 1970). The consistency of the calculation technique used to collect the data is measured by the reliability of the questionnaire. The findings of the survey and the outcomes of the six tested hypotheses are presented below.

### 4.6. Reliability Analysis

As we have used closed-ended questions with a five-degree Likert scale ranging from "strongly disagree" 1 to "strongly agree" 5, the respondents were asked to rate their level of agreement and confidence for this research, and the Cronbach alpha was used to assess the internal accuracy and reliability of the questionnaire items used in this study. It is a dependable tool that measures the reliability of several items in a designed questionnaire. The collected data are entered into SPSS to perform the reliability review. The value of the Cronbach alpha is commonly used to assess item reliability. The internal consistency between the items is good if the Cronbach alpha is equal to or greater than 0.70, and the data are considered accurate for further study. If the alpha value is lower than 0.5, then it shows poor internal consistency (George and Mallery 2003; Kline 2000).

The overall reliability in the Table 1 is 0.924, which shows good internal consistency. As per variable reliability in Table 2, learning and physical facilities ($\alpha$ = 0938), employability ($\alpha$ = 0.669), female participation ($\alpha$ = 0.906), skill development ($\alpha$ = 0.865), and teacher training ($\alpha$ = 0.925) are the projected acceptable values of the Cronbach alpha. The Cronbach alpha value of industrial linkage ($\alpha$ = 0.665) lies between 0.6–0.7, which shows

adequate internal consistency. For any primary and developmental research, the value of the Cronbach alpha is tolerable up to a value of 0.6 (Robinson et al. 1991).

**Table 1.** Overall reliability.

| Cronbach's Alpha | Sample Size |
|:---:|:---:|
| 0.924 | 500 |

Note: $0.9 \leq \alpha$ = Excellent, $0.8 \leq \alpha < 0.9$ = Good, $0.7 \leq \alpha < 0.8$ = Acceptable, $0.6 \leq \alpha < 0.7$ = Questionable, $0.5 \leq \alpha < 0.6$ = Poor, $\alpha < 0.5$ = Unacceptable.

**Table 2.** Variable reliability.

| Variables | Cronbach's Alpha | Sample Size |
|:---:|:---:|:---:|
| TVET | 0.903 | 500 |
| Learning and Physical Facilities | 0.942 | 500 |
| Employability | 0.669 | 500 |
| Female Participation | 0.905 | 500 |
| Skill Development | 0.865 | 500 |
| Industrial Linkage | 0.650 | 500 |
| Teacher Training | 0.925 | 500 |

With an $R^2$ of 0.546, the model's independent variables can account for 54.6% of the change in the dependent variable and explained 54% of the variation, according to the adjusted $R^2$ presented in Table 3.

**Table 3.** Model Summary.

| Model | R | R Square | Adjusted R Square | Std. Error of the Estimate |
|:---:|:---:|:---:|:---:|:---:|
| 1 | 0.739 [a] | 0.546 | 0.540 | 0.60414 |

[a] Predictors: (Constant), TT, FP, IL, EM, LPS, LS.

The standard error of the estimate in this model is 60.41 percent, which describes the standard deviation of the estimate (the factors in this model that could affect Technical Vocational Education and Training) (TVET).

The processed data, which are the population parameters, have a significant level of 0% in the ANOVA Table 4, indicating that the data are optimal for forming a conclusion on the population's parameters. It also shows that the model was statistically significant, with f (6493) = 98.759, $p < 0.001$, and $R^2$ = 0.546, and six problem elements, which operated as independent variables, had a considerable impact on the dependent variable.

**Table 4.** ANOVA results.

| Model | | Sum of Square | Df | Mean Square | F | Sig. |
|:---:|:---:|:---:|:---:|:---:|:---:|:---:|
| | Regression | 216.272 | 6 | 36.045 | 98.759 | 0.000 [b] |
| 1 | Residual | 179.936 | 493 | 0.365 | | |
| | Total | 396.208 | 499 | | | |

[a] Dependent Variable: TVET, [b] Predictors: (Constant), TT, FP, IL, EM, LPS, LS.

In the Table 5, with a beta coefficient of 0.404 and a sig. value of 0.000, the beta explains the role of each independent variable of the (LPS) learning and physical facilities in explaining TVET. The independent variables of (EM) employability ($\beta$ = 0.233; $p$ = 0.000), (FP) female participation ($\beta$ = 0.080; $p$ = 0.007), (LS) skill development ($\beta$ = 0.119; $p$ = 0.039), and (TT) teacher training ($\beta$ = 0.199; $p$ = 0.000), showed positive significant impact on Technical Vocational Education and Training, except (IL) for industrial linkage, where ($\beta$ = −0.096; $p$ = 0.075).

**Table 5.** Coefficients.

| Model | Unstandardized Coefficients | | Standardized Coefficients | T | Sig. |
|---|---|---|---|---|---|
| | B | Std. Error | Beta | | |
| (constant) | 0.102 | 0.198 | - | 0.512 | 0.609 |
| LPS | 0.404 | 0.042 | 0.424 | 9.537 | 0.000 |
| EM | 0.233 | 0.047 | 0.187 | 4.931 | 0.000 |
| FP | 0.080 | 0.029 | 0.088 | 2.708 | 0.007 |
| LS | 0.119 | 0.058 | 0.097 | 2.069 | 0.039 |
| IL | −0.096 | 0.054 | −0.069 | −1.785 | 0.075 |
| TT | 0.199 | 0.042 | 0.229 | 4.773 | 0.000 |

Dependent variable: TVET$_a$.

The regression analysis in Table 6 shows that there is a significant impact of the independent variable on the dependent variable Technical Vocational Education and Training. Moreover, the correlation analysis shows that the correlation between variables also follows. The Pearson correlation values were 0.674, 0.469, 0.550, 0.346, and 0.616 and were significant at the 0.01 levels, respectively. There was no correlation between TVET and FP; the Pearson correlation showed the value as 0.015, which was not significant. The results of the correlation and regression analysis support all the hypotheses, i.e., the factors have a strong and positive relation with the dependent variable, except for the correlation between TVET and FP.

**Table 6.** Correlation analysis.

| | Correlation | TVET | LPS | EM | FP | LS | IL |
|---|---|---|---|---|---|---|---|
| LPS | Pearson Correlation | 0.674 ** | | | | | |
| EM | Pearson Correlation | 0.469 ** | 0.400 ** | | | | |
| FP | Pearson Correlation | 0.015 | −0.154 ** | 0.195 ** | | | |
| LS | Pearson Correlation | 0.550 ** | 0.581 ** | 0.488 ** | 0.000 | | |
| IL | Pearson Correlation | 0.346 ** | 0.408 ** | 0.491 ** | 0.075 * | 0.552 ** | |
| TT | Pearson Correlation | 0.615 ** | 0.697 ** | 0.356 ** | −0.168 ** | 0.670 ** | 0.392 ** |

TVET = Technical Vocational Education and Training, LPS = learning and physical facilities, EM = employability, FP = female participation, LS = skill development, TT = teacher training, IL = industrial linkage. * $p < 0.05$; ** $p < 0.01$.

### 4.7. Hypothesis Testing

The dependent variable TVET was regressed on predicting the variables LPS, EM, FP, LS, TT, and IL to test the hypothesis (Table 7). LPS, EM, FP, LS, and TT significantly predicted TVET f $(6, 493) = 98.759$, $p < 0.001$, which indicates that TVET can play a significant role in shaping LPS (b = 0.404, $p = 0.000 < 0.001$), EM (b = 0.233, $p = 0.000 < 0.001$), FP (b = 0.080, $p = 0.007 < 0.05$), LS (b = 0.119, $p = 0.039 < 0.05$), and TT (b = 0.199, $p = 0.000 < 0.001$). These results clearly show the positive effect of the LPS, EM, FP, LS, and TT. Moreover, the $R^2 = 0.546$ shows that the model explains 54.6% of the variance in TVET. IL has a negative (b = −0.096) and an insignificant ($p = 0.075 > 0.05$) impact on TVET. Table 7 shows the summary of the findings.

**Table 7.** Hypothesis testing.

| Hypothesis | Regression Weights | Beta Coefficient | R2 | F | t, Value | p, Value | Hypothesis Supported |
|---|---|---|---|---|---|---|---|
| H$_1$ | LPS→TVET | 0.404 | 0.546 | 98.759 | 9.537 | 0.000 | Yes |
| H$_2$ | EM→TVET | 0.233 | 0.546 | 98.759 | 4.931 | 0.000 | Yes |
| H$_3$ | FP→TVET | 0.080 | 0.546 | 98.759 | 2.708 | 0.007 | Yes |
| H$_4$ | LS→TVET | 0.119 | 0.546 | 98.759 | 2.069 | 0.039 | Yes |
| H$_5$ | IL→TVET | −0.096 | 0.546 | 98.759 | −1.785 | 0.075 | No |
| H$_6$ | TT→TVET | 0.199 | 0.546 | 98.759 | 4.773 | 0.000 | Yes |

Note: $p < 0.05$ TVET: Technical Vocational Education and Training, LPS: Learning and Physical facilities, EM: Employability, FP: Female Participation, LS: Skill Development, TT: Teacher Training, IL: Industrial Link.

## 5. Discussion

The H1 *p*-value is 0.000 and less than the 0.05 confidence level; hence, the conclusion is that LPS has a significant effect on TVET. In the current study, 66% of the respondents identified the scarcity of materials. The findings about the learning and physical situation agreed with Adiviso (2011), Shah (2004), and Inamullah (Inamullah et al. 2009), who identified the scarcity of machinery, tools, and books and the bad condition of the available ones. It is widely acknowledged that the training environment should be a realistic representation of the actual workplace. The dearth and shortage of facility construction resources was also pointed out by Ahmed. A (Ahmed and Khan 2018).

In this research, 98% agreed that TVET could be helpful to find employment. In addition, the variable of employment has a statistically significant effect on TVET. A study by Gita Subrahmanyam found similar results, stating that TVET is widely acknowledged as having a crucial role to play in addressing youth unemployment. However, in order to hire young people, TVET institutions will need to undergo a significant technological transition (Subrahmanyam 2013). Qazi at al. emphasized that there is a unidirectional relationship between higher education and unemployment (Qazi et al. 2017), and Nugraha at al. consider employability skills essential for TVET graduates to gain employment (Nugraha et al. 2020). A study by Ullah. R has also confirmed that people with technical training have more employment opportunities in the job market and that it creates a better source of income, especially for those who have little or no access to higher education (Ullah et al. 2016). Our current findings, where 56% of the respondents strongly agree with the statement that TVET would facilitate more workers in CPEC through job creation, are also similar to those of M.M. Zia (Zia and Waqar 2018), who empirically examines the overall amount of employment created under six road development projects and the labor composition of Pakistani and Chinese nationals. According to the data gathered, just 7% of the total jobs produced are attributable to Chinese people, while 93% of workers are Pakistani, which ultimately means that the skilled workers obtained employment.

Female participation had a statistically significant effect on TVET. This result is consistent with those of Ayub, Kouser, Amor, Omer, Alam, and Forhad. Ayub, in her study on parental influence and student attitude on TVET, investigated the idea that cultural constraints, male dominance, and lack of counseling stop females from having equal possibilities to enter Pakistan's TVET (Ayub 2017). Similarly, Kouser (Parveen et al. 2020) conducted a study on trends of enrollment, and Farid (Safarmamad 2019), on the influential factors of student enrollment, discussed the idea that technical disciplines were overwhelmingly male and that young ladies were hesitant to pursue and shine in these areas. The scholars Amoor and Umar (2015) and Alam and Forhad (2021) discovered noteworthy findings about women students' participation in TVET, where the negative public impression and recognition of TVET as the last option of schooling, the low intelligence quotient, the low academic achievement, and the job instability were some of the primary issues contributing to women's low involvement in technical and vocational education. Student interviews on the assessment of the momentous factors resulting in the lower female participation in TVET by the International Rescue Committee (IRC) identified, in particular, training in a conventionally male industry; they revealed a lack of knowledge about TVET's strengths, financial interest, inadequate financial support, and uncertainties about future employment opportunities (United States Agency 2017). The findings of this research revealed that female participation is constrained for various reasons, such as school dropout (80% agree), social and cultural reasons (80% agree), and lack of guidance (72% agree).

Lack of skills had a statistically significant effect on TVET. This result is consistent with the results of Magsi, Amjad, and Anwar. Magsi (2016) noted that entering into the labor market without any skills and qualifications causes low income. Similarly, Amjad, (Amjad 2005) in his study on skill and competencies, highlighted the issue of the low-skill trap, and he emphasized that if Pakistan wants to enter into a skilled economy, it must get out of this trap; Anwar (Anwer and Gill 2020) suggested that communication skills provide higher

opportunities for career growth. Thomas Chamorro called skills the global currency, where individual skills may depreciate if they are not updated or used (Chamorro-Premuzic and Becky 2019). Ansari claims that low funding and less emphasis is causing challenges to skill development in TVET (Ansari and Wu 2012); however, Lall and Weiss (2003) suggested using technology skills to tackle new challenges.

Teacher training also had a statistically significant effect on TVET. Salleh (Salleh and Sulaiman 2020) and Pirzada.G (Pirzada 2020) suggested changing the ecosystem of TVET by prioritizing teacher training for new and in-service teachers for teacher development. However, according to Zamir (2019) and Memon (2007), the teachers' ability and confidence in bringing technology into their classrooms is dependent on their perceptions and their readiness to accept new technology.

## 6. Conclusions and Recommendations

This study, which investigated six different variables, potentially influences the TVET in Pakistan. The results reveal that learning and physical facilities, employability, female participation, skill development, and teacher training are five variables that influence technical vocational education in Pakistan; however, industrial linkage does not have a significant effect on TVET. Pearson's correlation was used to test the possible relationship among the variables, and a regression analysis identified the variables that were significant.

Based on the findings, most TVET institutions have poor learning quality and infrastructure. Insufficient facilities and educational infrastructure hinder skill development and employment. Male dominance and TVET are cultural restrictions. Female work and institution participation are improving, but not by enough. Every industry, including CPEC, has few female workers. Mismatched TVET and industry diminish TVET's contribution to industry. Teachers and trainees lack modern methodological and technological abilities. This shows the professors' and the trainees' CPEC ignorance. The findings have offered evidence of the existing limitations in Pakistan's TVET, which are influencing not only the quality of the workers, but also the growth of the CPEC projects directly and indirectly. As a result, it appears that TVET institutions lack the necessary human and physical possessions for backing their learning/training programs. As a consequence, they do not create enough skilled graduates to meet the labor market demands. Therefore, rather than focusing on the quantity of TVET expansion, attention should be paid to the quality of it. Despite the fact that the structure has been altered and is undergoing leapfrog growth, more students are gaining access to education. However, due to unexpected economic changes (CPEC), more skilled workers are needed. As a result, the advantages of TVET in this new context should be on a macro basis, and the educational method should be based on the lifelong learning principle.

**Author Contributions:** N.B. completed the methodology, the preparation of the questionnaire, the SPSS application, the data integration, and the manuscript writing. S.Y. helped in research design, supervision, and expert opinion. E.A. helped with the SPSS and the questionnaire distribution and filling. All authors have read and agreed to the published version of the manuscript.

**Funding:** This research received no external funding.

**Institutional Review Board Statement:** Not applicable.

**Informed Consent Statement:** Not applicable.

**Data Availability Statement:** Not applicable.

**Conflicts of Interest:** All the listed authors declare that they do not have any conflict of interest.

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
