# Peer review of "Emerging Challenges in Technical Vocational Education and Training of Pakistan in the Context of CPEC"

_economies, doi:10.3390/economies10070153_

Round 1
Reviewer 1 Report
# The background to the study is clearly presented.
# Some of the conclusions drawn from the content presented in sections 2.2, 2.3 and 2.4 do provide data or evidence to support these. For instance in section 2.2, there are claims that there is a growing young population and that participation rate is low and so there is high unemployment and this affects health, education and quality of life. The connections between these claims is unfounded. Besides, it is not clear how all this relates to the main argument in the article. Similiarly, in section 2.4, the connection between the quality of TVET colleges, communication between workplaces and TVET systems etc. is unclear. In these sections, bits of information is presented in an incoherent manner. The line of argument is not clear.
I am not qualified to comment on the statistics. The data analysis section is too descriptive. Connections between the analysis and the main argument in the paper need to be stated.
The conclusion section needs to reflect the findings. The title suggests some recommendations are included but there are norecommendations as such.
There are several editorial errors in relation to grammar and sentence structures.
Author Response
Reviewer 1
Comments;
Some of the conclusions drawn from the content presented in sections 2.2, 2.3 and 2.4 do provide data or evidence to support these. For instance, in section 2.2, there are claims that there is a growing young population and that participation rate is low and so there is high unemployment and this affects health, education and quality of life. The connections between these claims are unfounded. Besides, it is not clear how all this relates to the main argument in the article. Similarly, in section 2.4, the connection between the quality of TVET colleges, communication between workplaces and TVET systems etc. is unclear. In these sections, bits of information are presented in an incoherent manner. The line of argument is not clear.
I am not qualified to comment on the statistics. The data analysis section is too descriptive. Connections between the analysis and the main argument in the paper need to be stated.
The conclusion section needs to reflect the findings. The title suggests some recommendations are included but there are no recommendations as such.
There are several editorial errors in relation to grammar and sentence structures.
Answers:
- As per your suggestion about section 2.2 that “the claims and connections of the given statement at the end of the part is not linked with the main argument” therefore the mentioned lines (125-127) have been removed.
- To present the information in an articulate manner and to make argument clearer, the mentioned paragraph of the section 2.4 is revised, and more authenticated references have been added to make our argument stronger.
- The Data analysis section which was pointed as to be more descriptive, has been made modified as short and précised.
- By agreeing with your kind suggestion, the conclusion part is revised and the main findings of the research has been added accordingly.
- The whole paper is revisited again and removed the grammatical errors and also corrected the sentence structures.

Reviewer 2 Report
The paper "Emerging Challenges in Technical Vocational Education and Training of Pakistan in the context of CPEC " basically concerns the evaluation of the quality of TVET vocational training system directed to prepare the Pakistani workforce. As the authors state in the introduction, Pakistani firms lack of qualified personnel has been attributed to the inadequacy of TVET t. The paper seeks to test whether this hypothesis is true and what factors have led to this inadequacy.
The paper is well written , the literature review is comprehensive , the logical scansion of arguments is acceptable but there are some big methodological weaknesses that need to be resolved for accepting the paper.
(a) Sample and target population
The sample is composed by 500 people composed by 320 students, 70 teachers, 70 technical job holders, 20 entrepreneurs, and 20 others. They filled the online Google form and all the responses were first recorded and then were analyzed according to the received data.
The target population for this research is a”ll the concerned people related to TVET 2 Pakistan i.e. teachers, students, technical job holders, entrepreneurs and other relevant people from the Govt. and private TVET institutions in Pakistan.”
There are several issues regarding the link between sample and target population
1 The target population is defined in a totally generic way: talking about teachers, students, technical job holders, entrepreneurs and other relevant people from the Govt. and private TVET institutions is totally insufficient to define this population Especially since the reader cannot know what the acronym Govt means that is not made explicit, and which private Pakistani institutions are connected to the TVTET.
2 What is the numerical size of the subpopulations of teachers, students, technical job holders, entrepreneurs and other relevant people belonging to the target population? We need this information to understand whether the numerical size of the different components of the sample is adequate
3 How did the authors select the sample of individuals participating in the survey? Is it a random sample or not?
4 If it is observational in nature what are the criteria for the selection? Was it chosen by the authors of the research or is it composed of the individuals who freely agreed to participate in the survey ?
5 If the sample is observational what analysis was done to rule out that it is affected by self-selection?
B Variables
The authors commendably and comprehensively analyze the literature to understand what factors influence TVTET coming to identify Learning and physical facilities, Employment opportunities, , Female participation, Education and Training, Skills for self-employment , Industrial Link, : Teachers Training. They all have a positive and significant impact on TVET. However:
1. Although the literature cited indicates these independent variables and TVTET as quantitative variables in the paper they are derived by responses to questionnaire expressed as five-degree Likert-scale ranged from strongly disagree to strongly agree. Now, it is completely different to deal with quantitative indicators and subjective judgments of people with respect to them. How the authors could reconcile the two approaches?
2. In the paper appropriately a Reliability Analysis is done using Cronbach's Alpha to assess Internal consistency between items and of the questionnaire in general. But nothing is said about the kind of questions. Thus the readers cannot judge the validity of questions and answers used to interpret the phenomena studied.
3. In the analysis of the results the authors alternate the terms Employment, Unemployment, Skills and Lack of Skills in a totally confusing way. Do they analyze the effects on TVET of Employment, and Skills or those of Unemployment and Lack of Skills , as it seems from the explanation? And in this second case why the coefficients are positive instead of negative?
C Regression
The authors utilize a linear regression, to perform their analysis. However at least some of the variables such as Learning and physical facilities, Education and Training, Teachers Training are “Treatments” that differ from the other usual independent variables Therefore to get acceptable results it is necessary a Causal Inference methodology comparing subpopulations subjected to the Treatment itself and control subpopulations not subjected to the Treatment ( example Matching, Difference in difference).
D Discussions and Conclusions
While discussions are complete and full of references to the literature the very brief conclusions are wholly inadequate. In particular the policy remarks about "the government's primary focus was on the quantitative components, with little attention paid to the qualitative aspects" are misplaced. In fact, in the paper, there are no quantitative tables or previous considerations to analyze the objective qualitative level of factors. All the analyses and remarks are based on subjective opinions of people participating to the survey. Moreover the reader does not know if the questions composing the questionnaire concern some qualitative aspect of factors that determine TVET. Therefore the conclusions do not fulfill the purpose of the paper outlined in the abstract and the introduction
Author Response
Reviews and Answers.
The paper "Emerging Challenges in Technical Vocational Education and Training of Pakistan in the context of CPEC " basically concerns the evaluation of the quality of TVET vocational training system directed to prepare the Pakistani workforce. As the authors state in the introduction, Pakistani firm’s lack of qualified personnel has been attributed to the inadequacy of TVET t. The paper seeks to test whether this hypothesis is true and what factors have led to this inadequacy.
The paper is well written, the literature review is comprehensive, the logical scansion of arguments is acceptable but there are some big methodological weaknesses that need to be resolved for accepting the paper.
(a) Sample and target population
The sample is composed by 500 people composed by 320 students, 70 teachers, 70 technical job holders, 20 entrepreneurs, and 20 others. They filled the online Google form and all the responses were first recorded and then were analyzed according to the received data.
The target population for this research is a”ll the concerned people related to TVET 2 Pakistan i.e. teachers, students, technical job holders, entrepreneurs and other relevant people from the Govt. and private TVET institutions in Pakistan.”
There are several issues regarding the link between sample and target population
Comment 1:
The target population is defined in a totally generic way: talking about teachers, students, technical job holders, entrepreneurs and other relevant people from the Govt. and private TVET institutions is totally insufficient to define this population Especially since the reader cannot know what the acronym Govt means that is not made explicit, and which private Pakistani institutions are connected to the TVTET.
Answer.
The targeted population involved in this study including staffs (Teachers, technical job, entrepreneurs) and the students of the specific institutions with located cities are identified properly to define the population in the part 4.3.
Comment 2:
What is the numerical size of the subpopulations of teachers, students, technical job holders, entrepreneurs and other relevant people belonging to the target population? We need this information to understand whether the numerical size of the different components of the sample is adequate
Answer:
The whole section 4.3 is revised as per your suggestion and recommendations. The numerical size of the sub-population of staffs (Teachers, technical job, entrepreneurs) and the students has been identified.
Comment 3:
How did the authors select the sample of individuals participating in the survey? Is it a random sample or not?
Comment 4:
If it is observational in nature what are the criteria for the selection? Was it chosen by the authors of the research or is it composed of the individuals who freely agreed to participate in the survey?
Comment 5:
If the sample is observational what analysis was done to rule out that it is affected by self-selection?
Answers of comment 3-5:
The answers of the comment 3, comment 4, and comment 5 has been revised and updated. The mentioned section has been revisited, and the concerned paragraph in section 4.2 is modified according to the identified issues.
B Variables
The authors commendably and comprehensively analyze the literature to understand what factors influence TVET coming to identify Learning and physical facilities, Employment opportunities, , Female participation, Education and Training, Skills for self-employment , Industrial Link, : Teachers Training. They all have a positive and significant impact on TVET. However:
Comment 1:
Although the literature cited indicates these independent variables and TVTET as quantitative variables in the paper they are derived by responses to questionnaire expressed as five-degree Likert-scale ranged from strongly disagree to strongly agree. Now, it is completely different to deal with quantitative indicators and subjective judgments of people with respect to them. How the authors could reconcile the two approaches?
Answer:
Quantitative indicators and subjective judgments both have their strengths and weaknesses, because sometimes narratives and sometimes numbers are more useful. Oftentimes, a mix of quantitative and qualitative data provides the most useful information. However, arriving at the correct-manageable mixture of them is not an easy task, this is why we have used a mixed approach. In both, quantitative indicators and subjective judgments, the key research statement is the research objective itself, so the aim was to use qualitative approach to explain quantitative results.
Comment 2.
In the paper appropriately a Reliability Analysis is done using Cronbach's Alpha to assess Internal consistency between items and of the questionnaire in general. But nothing is said about the kind of questions. Thus, the readers cannot judge the validity of questions and answers used to interpret the phenomena studied.
Answer:
Cronbach's alpha is the most common measure of internal consistency ("reliability"). It is used when researches have multiple Likert questions in any survey/questionnaire that form a scale, and wish to determine if the scale is reliable or not. Details about the kinds of questions have been added.
Comment 3:
In the analysis of the results the authors alternate the terms Employment, Unemployment, Skills and Lack of Skills in a totally confusing way. Do they analyze the effects on TVET of Employment, and Skills or those of Unemployment and Lack of Skills, as it seems from the explanation? And in this second case why the coefficients are positive instead of negative?
Answer:
The alternative terms like unemployment and lack of skills have been replaced by employability and skill development as per conceptual framework.
C Regression
The authors utilize a linear regression, to perform their analysis. However at least some of the variables such as Learning and physical facilities, Education and Training, Teachers Training are “Treatments” that differ from the other usual independent variables Therefore to get acceptable results it is necessary a Causal Inference methodology comparing subpopulations subjected to the Treatment itself and control subpopulations not subjected to the Treatment ( example Matching, Difference in difference).
Answer:
This study is exploratory and descriptive in nature where the data was collected through online survey. Because the nature of the study is not experimental so there are no experimental and control groups in the study and the only way for a research method to determine causality is through a properly controlled experiment. Moreover the major purpose/aim of the study was to explore the major factor influencing technical vocational education and training and relationship between variables like learning facilities, skill development, employability, female participation and industrial linkage. Therefore, the main focus was on finding correlation and regression rather casual inference.
D Discussions and Conclusions
While discussions are complete and full of references to the literature the very brief conclusions are wholly inadequate. In particular the policy remarks about "the government's primary focus was on the quantitative components, with little attention paid to the qualitative aspects" are misplaced. In fact, in the paper, there are no quantitative tables or previous considerations to analyze the objective qualitative level of factors. All the analyses and remarks are based on subjective opinions of people participating to the survey. Moreover, the reader does not know if the questions composing the questionnaire concern some qualitative aspect of factors that determine TVET. Therefore the conclusions do not fulfill the purpose of the paper outlined in the abstract and the introduction.
Answer:
The conclusions section (6) has completely revisited and modified as per your suggestions and recommendations. However as per the suggestion of the other reviewer, the study findings have been added to the conclusion part.

Round 2
Reviewer 2 Report
The authors have made all requested corrections and improvements
(a) Sample and target population
1.The targeted population involved in this study including staffs (Teachers, technical job, entrepreneurs) and the students of the specific institutions with located cities are identified properly to define the population in the part 4.3.
2.The whole section 4.3 is revised as per your suggestion and recommendations. The numerical size of the sub-population of staffs (Teachers, technical job, entrepreneurs) and the students has been identified.
3 The answers of the comment 3, comment 4, and comment 5 has been revised and updated. The mentioned section has been revisited, and the concerned paragraph in section 4.2 is modified according to the identified issues.
New comment
The new par 4.2 and 4.3 are adequate
B Variables
1 Quantitative indicators and subjective judgments both have their strengths and weaknesses, because sometimes narratives and sometimes numbers are more useful. Oftentimes, a mix of quantitative and qualitative data provides the most useful information. However, arriving at the correct-manageable mixture of them is not an easy task, this is why we have used a mixed approach. In both, quantitative indicators and subjective judgments, the key research statement is the research objective itself, so the aim was to use qualitative approach to explain quantitative results.
2.Cronbach's alpha is the most common measure of internal consistency ("reliability"). It is used when researches have multiple Likert questions in any survey/questionnaire that form a scale, and wish to determine if the scale is reliable or not. Details about the kinds of questions have been added.
3 The alternative terms like unemployment and lack of skills have been replaced by employability and skill development as per conceptual framework.
New comment
The corrections are satisfactory
Regression
1.This study is exploratory and descriptive in nature where the data was collected through online survey. Because the nature of the study is not experimental so there are no experimental and control groups in the study and the only way for a research method to determine causality is through a properly controlled experiment. Moreover the major purpose/aim of the study was to explore the major factor influencing technical vocational education and training and relationship between variables like learning facilities, skill development, employability, female participation and industrial linkage. Therefore, the main focus was on finding correlation and regression rather casual inference.
New comment
The explanations answer the previous comment
D Discussions and Conclusions
1. The conclusions section (6) has completely revisited and modified as per your suggestions and recommendations. However as per the suggestion of the other reviewer, the study findings have been added to the conclusion part.
New comment
The modifications of the discussion and the conclusions are now useful to understand the paper issue